# Analysis of Kinematic Variables According to Menstrual Cycle Phase and Running Intensity: Implications for Training Female Athletes

Carolina Domínguez-Muñoz [1,*], Juan del Campo [1], Alberto García [2], José Guzmán [3], Rafael Martínez-Gallego [3] and Jesús Ramón-Llin [4]

1 Department of Physical Education, Sport and Human Motricity, Autonomous University of Madrid, 28049 Madrid, Spain; juan.delcampo@uam.es
2 Department of Sports, Faculty of Physical Activity and Sports Sciences (INEF), Polytechnic University of Madrid, 28040 Madrid, Spain; agarbasn1@me.com
3 Department of Physical and Sports Education, University of Valencia, 46010 Valencia, Spain; jose.f.guzman@uv.es (J.G.); rafael.martinez-gallego@uv.es (R.M.-G.)
4 Department of Teaching of Physical Education, Artistic and Music, University of Valencia, 46010 Valencia, Spain; jesus.ramon@uv.es
* Correspondence: mastrainingpro@gmail.com

**Abstract:** Depending on the phase of the menstrual cycle, different values of running kinematic variables can be obtained. The aim of this study is to analyze whether there are changes in the kinematic variables in running throughout the menstrual cycle and to relate them to running performance and injury prevention. Eight regular female runners and triathletes performed a maximal treadmill test, as well as a submaximal test (6′ stages at 50%, 60% and 80% of maximal aerobic speed) in each of the phases of the menstrual cycle: menstruation phase (day $2.4 \pm 0.7$), follicular phase (day $10.4 \pm 2.2$) and luteal phase (day $21.8 \pm 2.1$). Running dynamics were measured using RunScribe. For parametric data, a general linear model of repeated measures was applied, with two intrasubject independent variables, menstrual cycle phases (with three levels: Menstruation, Follicular, and Luteal) and running intensity (with four levels relative to the maximum speed reached in the test: 100%, 80%, 60%, and 50%). For variables with non-normal distributions, Friedman tests were performed with Wilcoxon post-tests adjusted for significance according to Bonferroni. The maximum stance velocity from foot strike to the point of maximum pronation (°/s) was higher in the menstruation phase than in the follicular and luteal phases ($p = 0.008$), the step rate (s/min) was higher in the follicular phase than in the menstruation and luteal phases ($p = 0.049$), the vertical velocity (m/s) was lower in the follicular phase than in the menstruation ($p = 0.004$) and luteal phases ($p = 0.003$), and the contact time (ms) was lower in the luteal phase than in the menstruation and follicular phases. These results suggest that training at high intensities could be a factor in greater risk of injury in female athletes, especially in the menstruation phase, finding in the luteal phase and at an intensity of 80% a greater efficiency in the running.

**Keywords:** menstrual cycle; running; kinematic variables; injury; running intensity



## 1. Introduction

Kinematic variables, such as stride and step length (m), stride angle, step rate (s/min), vertical speed (m/s), contact time (ms), contact ratio (%), flight time (ms), and flight ratio (%), have significant correlations with performance (Björklund et al., 2019 [1]; Danion et al., 2003 [2]; Santos-Concejero et al., 2014 [3]). Increasing running speed leads to reduced contact time, increased stride length, shorter stance phase, and longer flight time (Orendurff et al., 2018 [4]; Albiach et al., 2021 [5]). Additionally, Orendurff et al. (2018) [4] found increased dorsal flexion at initial contact and ankle plantar flexion during the early leg swing. Increased running speed can modify some kinematic variables in men that are

also shared with women; specifically, Gijon-Nogueron et al. (2019) [6] investigated female runners at increased speeds and observed an increase in flight time, decrease in contact time and alterations in the initial contact, stance, and propulsion phases.

Increased running speed and/or prolonged running over time will cause signs of fatigue. Fatigue studies in the running indicate that contact time may increase during the stance phase along with ankle, knee, and hip range of motion (angular moments) (Mohler et al., 2021 [7]; Mei et al., 2019 [8]). However, step rate and stride length remain relatively unchanged in trained runners experiencing fatigue (Mohler et al., 2021 [7]), while Cartón-Llorente et al. (2022) [9] noted a slight increase in step rate among trained runners. Li et al. (2022) [10] studied fatigue between a 5 km run and a 10 km run and observed a decline in propulsion and increased foot eversion during stance, particularly after 10 km. They also noted greater dorsal flexion at 5 km compared to 10 km and reduced plantar flexion after both distances. Increased fatigue during running causes changes in some kinematic variables in male runners, but what these changes would look like in female runners has not been described.

Changes in kinematic variables due to fatigue may vary between novice and experienced runners. Novice runners exhibit increased stride width variability for performance and stability, while elite runners demonstrate shorter stride length and width (Panday et al., 2022 [11]). Novices struggle with impact absorption during the stance phase, while both novice and elite runners adjust stride length to improve economy, reduce ankle stiffness, and increase knee stiffness, thereby decreasing oxygen consumption. These studies do not differentiate between men and women.

Some changes in kinematic variables may be related to injury, and knowing their relationship may help prevent injury. Injury prevention research has linked increased foot pronation during running fatigue to an increased risk of injury (Mei et al., 2019 [8]; Bramah et al., 2018 [12]). Injured runners show increased knee extension, ankle dorsiflexion, trunk tilt, contralateral pelvic drop and hip adduction during the stance phase, with contralateral pelvic drop being a significant kinematic parameter associated with injury (Bramah et al., 2018 [12]). Hip adduction can be an injury risk for both men and women, with women having greater hip adduction associated with specific injuries. Higher weight, longer flight time, lower stride frequency, and greater training at higher speeds also contribute to injury in female runners (Winter et al., 2020 [13]).

Wilhoite et al. (2021) [14] found that simultaneous knee flexion at stance and foot eversion help reduce impact and prevent injury. Runners experiencing lower limb discomfort tend to decrease stride length and increase contact time, negatively impacting the economy, increasing oxygen transport costs, and impairing impact absorption (Koldenhoven et al., 2020 [15]). Increased pronation is also associated with these runners. Another kinematic parameter in male runners with lower limb discomfort shared with female runners that increases the likelihood of injury is the decreased ability to attenuate impact.

Kiernan et al. (2018) [16] investigated injury risk prediction and discovered that injured runners experience higher vertical reaction forces and cumulative loads. In recreational runners, the number of strides per training session may correlate with the likelihood of injury. Male runners who are prone to injury have similarities with female runners and experience a greater impact on the tibia and sacrum, as well as a higher rate of vertical ground reaction force, potentially leading to a higher propensity for injury due to reduced impact absorption (Burke et al., 2022 [17]). Additionally, Popp et al. (2022) [18] focused on bone stress injuries in women, highlighting increased vertical load rate, vertical stiffness, and tibial impingement as contributing factors, while Wilzman et al. (2022) [19] analyzed plantar pressures in female runners to predict bone stress injuries, linking increased pressure on the first metatarsal with forefoot valgus, increased power, and a shift in load to the forefoot.

No studies have been located in the literature that relate the influence of the menstrual cycle on the kinematic variables of running, but there is literature that describes cutting and landing kinematics with an ACL injury risk perspective according to the different phases

of the menstrual cycle, placing the menstruation and ovulation phases at the highest risk of injury due to greater vertical ground reaction forces, tibial internal rotation at initial contact and hip internal rotation, accompanied by increased anterior knee laxity (Bell et al., 2014 [20]), and place the mid-luteal phase at the lowest risk of injury by locating a lower maximum valgus angle (Bingzheng et al., 2023 [21]).

Moreover, other running technique variables, such as swing excursion, maximum pronation velocity, maximum stance velocity from foot strike to the point of maximum pronation, maximum stance velocity from the point of maximum pronation to toe-off, time from the point of maximum pronation to toe-off, time from toe-off to the minimum swing point, and time from the maximum swing point to foot strike, have received limited analysis but may influence performance.

Studying the biomechanics of running, specifically through kinematic variables, provides coaches with insights into athletes' strengths and weaknesses (Jaén-Carrillo et al., 2020 [22]). This research aims to identify the running technique variables related to performance and influenced by the menstrual cycle and running intensity.

## 2. Materials and Methods

### 2.1. Subjects

Eight women, aged 37.1 ± 3.5 years, were recruited for this study. Inclusion criteria included a minimum of 3 years of regular running experience, training at least three times a week for 1 h, regular menstrual cycles (length of each cycle varies between approximately 24 and 38 days), no current use of contraceptives (with a minimum of 6 months since discontinuation), and no existing pathologies, illnesses, or injuries.

All participants received comprehensive information regarding the study's objectives and procedures and provided informed consent. They also signed a consent form allowing the use of photographic and/or audiovisual material, adhering to the ethical principles outlined in the World Medical Association's Declaration of Helsinki (2013). The study obtained approval from the Research Ethics Committee.

### 2.2. Design

Testing sessions were conducted individually between November 2021 and April 2022, considering the different phases of each participant's menstrual cycle.

Participants were instructed to maintain their regular diet and abstain from vigorous physical activity for 24–48 h before the test. They were also advised to avoid consuming food, stimulants, or ergogenic aids for at least three hours prior to the test.

Both a maximal and a submaximal test were performed by each participant on a treadmill (Viasys LE 200 CE, Bimedis, Kissimmee, FL, USA) inclined at 1% to replicate outdoor running conditions (Jones et al., 1996 [23]). Participants wore their usual training and/or competition shoes to minimize technical alterations.

### 2.3. Materials and Measurements

Each woman underwent testing at three different points in her menstrual cycle: (1) menstruation phase (day 2.4 ± 0.7); (2) follicular phase (day 10.4 ± 2.2); and (3) luteal phase (day 21.8 ± 2.1). Within each phase, participants completed a maximal test and a submaximal test 48–72 h later.

The start of the maximal test was 5–6 km/h slower than each participant's (most recent) best 10 km run time. This was an incremental test where every 1 min, the speed was increased by 1 km/h. The treadmill was always at a 1% gradient. The test ended when the participant could not maintain the speed. The maximal test provided kinematic variables of running (Albiach et al., 2021 [5]).

The submaximal test consisted of three intensities, each lasting 6 min: (1) 50% of the maximal speed achieved in the maximal test; (2) 60% of the maximal speed achieved in the maximal test; and (3) 80% of the maximal speed achieved in the maximal test. The treadmill

was always at a 1% gradient. This test enabled the collection of kinematic variables at each intensity level during running (Gijon-Nogueron et al., 2019 [6]).

The timing of the menstrual cycle phases was tracked using the Clue app (https://helloclue.com/), which was installed on each participant's mobile phone. The app allowed participants to sync their menstrual cycle information with the lead researcher, facilitating accurate tracking. Clue is a period-tracking app and a science-based menstrual and reproductive health resource with a database of 378,000 users and 4.9 million natural cycles. By utilizing self-reported tracking data, Clue can reveal statistically significant associations between cycle length variability and self-reported qualitative symptoms (Urteaga et al., 2021 [24]; Li et al., 2022 [10]).

The RunScribe system, consisting of two pods (one on each foot), was used for continuous analysis of kinematic variables during running. RunScribe is a validated running analysis platform that captures data from both feet at every stride, providing a comprehensive view of running mechanics (Koldenhoven et al., 2020 [15]; DeJong et al., 2020 [25]; Napier et al., 2021 [26]). The RunScribe Red Gait Lab (Moss Beach, CA, USA) makes it easy to analyze, from per-step CSV to full 500 Hz RawData IMU data. Using the RunScribe mobile app (version 3.4.0 (470) © 2021 RunScribe), calibration was performed automatically for each of the women before each measurement and each measurement was recorded and then analyzed in more detail on the RunScribe website.

### 2.4. Variables Analyzed

These are the kinematic variables that were analyzed by RunScribe Red Gait Lab:

Distances and angles. (1) Stride length: the distance between two successive placements of the same foot, consisting of two-step lengths. It is measured in meters. (2) Stride angle: the angle of the tangent of the parabola derived from the movement of a stride, formed by the stride length and height to which the foot rises. It is measured in degrees. (3) Swing excursion: the angle between the rearmost swing of the foot and the forwardmost swing of the same foot. It is measured in degrees. (4) Step length: the individual step length from right foot to left foot and from left foot to right foot. It is measured in meters.

Speeds and frequencies. (1) Maximum speed: the maximum speed reached in the maximum test. The start of the test was 5–6 km/h slower than his (most recent) best 10 km run time. It is an incremental test where every 1 min, the speed is increased by 1 km/h. The treadmill was always at a 1% gradient. It is measured in kilometers per hour. (2) Maximum pronation velocity: the maximum angular rate at which the foot pronates between the foot strike and the point of maximum pronation. It is measured in degrees per second. (3) Maximum stance velocity from the foot strike to the point of maximum pronation: the maximum speed of the foot during the damping phase from the foot strike to the point of maximum pronation. It is measured in degrees per second. (4) Maximum stance velocity from the point of maximum pronation to toe-off: the maximum speed of the foot during the propulsion phase from the point of maximum pronation to toe-off. It is measured in degrees per second. (5) Step rate: the number of steps a runner takes per minute. It is measured in steps per minute. (6) Vertical speed: the vertical speed of the foot at stance. It is measured in meters per second.

Times and percentages. (1) Contact time: the time that the foot is in contact with the ground. It is measured from heel to toe. It is measured in milliseconds. (2) Contact ratio: the percentage of time the feet are in contact with the ground. It is measured in percentage. (3) Flight time: the time that both feet are in the air. It is measured in milliseconds. (4) Flight ratio: the percentage of time the feet are in the air (flight). It is measured in percentage. (5) Time from the point of maximum pronation to toe-off: the time it takes for the foot to move from the point of maximum pronation to toe-off. It is measured in milliseconds. (6) Time from toe-off to min swing point: the time it takes for the foot to move from take-off to the rearmost swing point. It is measured in milliseconds. (7) Time from the max swing point to foot strike: the time it takes for the foot to move from the forward swing point to foot strike. It is measured in milliseconds.

### 2.5. Statistical Analysis

Statistical analysis was performed using SPSS 28.0 statistical software (IBM; Chicago, IL, USA). To select the kinematic variables provided by RunScribe, bivariate Spearman correlations were performed beforehand, and only variables with significant relationships with the maximum speed of the test were included. Correlations between the dependent variables and maximum speed were carried out using Pearson's R for continuous and normally distributed variables or Spearman's Rho. Mean (M) and standard deviation (SD) or median (Mn) and interquartile range (IQ) were used as descriptive statistics. K-S or Shapiro–Wilks tests for normality and Mauchly's test for sphericity were performed beforehand. To analyze the effect of menstrual cycle phases and test intensity on variables with a normal distribution, a general linear model of repeated measures was applied, with two intrasubject independent variables, menstrual cycle phases (with three levels: Menstruation, Follicular, and Luteal) and running intensity (with four levels relative to the maximum speed reached in the test: 100%, 80%, 60%, and 50%). The tests used the F statistic of assumed sphericity or Greenhouse–Geisser with post-hoc tests with Bonferroni adjustment. For variables with non-normal distributions, Friedman tests were performed with Wilcoxon post-tests adjusted for significance according to Bonferroni. Statistical significance was adjusted for *p*-values lower than 0.05. As effect size statistics, eta squared was used, with values of 0.01 small, 0.06 medium, and 0.14 large, while for the Wilcoxon tests, the r statistic was used, with values of 0.5 large, 0.3 medium, and 0.3 small.

## 3. Results

### 3.1. Correlations of Running Kinematics Variables with Maximal Velocity

Table 1 shows the results of the variables selected from RusScribe due to their significant correlation with maximum running speed.

**Table 1.** Correlations of running kinematics variables with maximal velocity.

| Kinematic Variables | Correlation with V max. | 95% C.I | |
|---|---|---|---|
| | | Lower C.I. | Upper C.I. |
| **Distance and angles** | | | |
| Stride Length (m) | 0.648 *** | 0.510 | 0.754 |
| Stride Angle (°) | 0.302 ** | 0.102 | 0.479 |
| Swing Excursion (°) | 0.599 ** | 0.453 | 0.714 |
| Step Length (m) | 0.668 *** | 0.536 | 0.768 |
| **Velocity and frequency** | | | |
| Maximal Speed (km/h) | 1 *** | 1 | 1 |
| Max Pronation Velocity (°/s) | 0.448 ** | 0.271 | 0.595 |
| Max Stance Velocity (FS-MP) (°/s) | 0.423 ** | 0.238 | 0.579 |
| Max Stance Velocity (MP-TO) (°/s) | 0.686 ** | 0.559 | 0.782 |
| Step rate | 0.556 ** | 0.395 | 0.684 |
| Vertical Speed (m/s) | −0.491 *** | −0.633 | −0.317 |
| **Times** | | | |
| Contact Time (ms) | 0.649 ** | −0.755 | −0.512 |
| Contact Ratio (%) | −0.536 ** | −0.668 | −0.370 |
| Flight Time (ms) | 0.490 ** | 0.335 | 0.645 |
| Flight Ratio (%) | 0.532 *** | 0.364 | 0.667 |
| Time (MP-TO) (ms) | −0.677 *** | −0.775 | −0.547 |
| Time (TO-Min Swing) (ms) | 0.442 *** | 0.265 | 0.590 |
| Time (Max Swing-FS) (ms) | 0.538 ** | 0.373 | 0.670 |

Max_stance_vel_MP—Maximum stance velocity from the point of maximum pronation to toe-off (°/s); Max_stance_vel_FS—Maximum stance velocity from the foot strike to the point of maximum pronation(°/s); Time_MP-TO—Time from the point of maximum pronation to toe-off (ms); Time_TO_min_swing—Time from toe-off to minimum swing point (ms); Time_TO_max_swing—Time from the maximum swing point to the foot strike (ms). *** = *p* < 0.001; ** = *p* < 0.01.

### 3.2. Test Performance

Maximum Speed (Max_V)

There were significant differences in the maximum speed achieved in the test depending on the phase of the menstrual cycle ($X_2^2 = 9.5$; $p = 0.009$; $\eta^2 = 0.16$), with the median being significantly higher in the follicular phase (Mn = 10. 6; IQ = 5.7) than in the luteal phase (Mn = 10.0; IQ = 4.9) (r = 0.31; $p = 0.002$) and with a significantly higher tendency than in the menstruation phase (Mn = 10.5; IQ = 5.7) (r = 0.22; $p = 0.031$) (Figure 1).

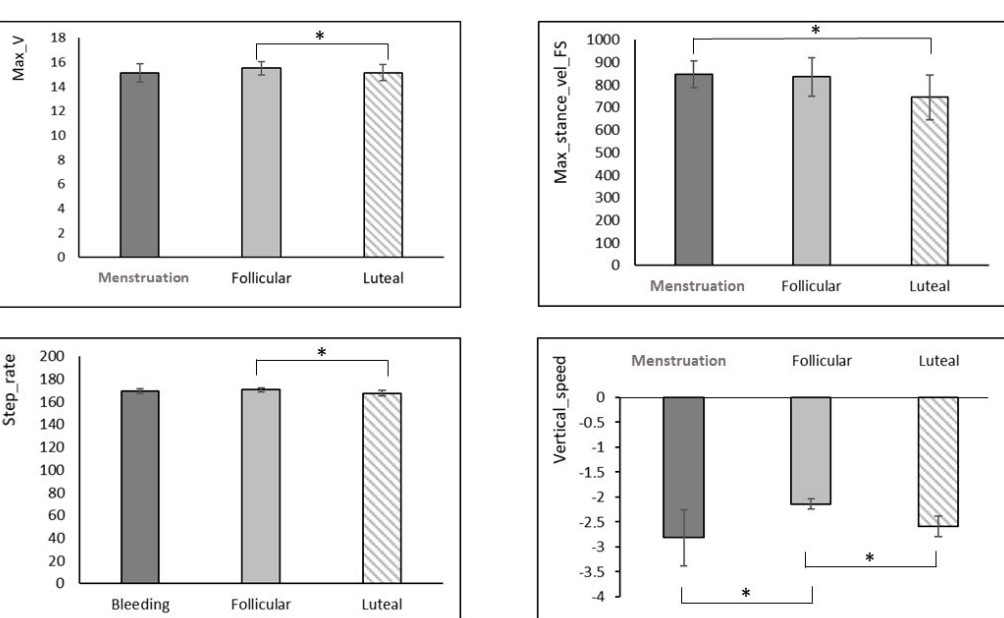

**Figure 1.** Comparison of the significant effect of the menstrual cycle phases on kinematic variables of velocity and frequency. Max_V—Maximum speed in test (m/s); Max_stance_vel_FS—Maximum stance velocity from the foot strike to the point of maximum pronation (°/s); Step_rate (steps/min); Vertical speed (m/s); * = $p < 0.05$.

### 3.3. Distances and Angles

3.3.1. Stride Length

The menstrual cycle phases had no significant effect on stride length ($F_2 = 0.298$; $p = 0.747$, $\eta^2 = 0.041$), which was slightly lower in the follicular phase (Table 2). Running intensity had a significant effect on stride length ($F_3 = 18.5$; $p < 0.001$, $\eta^2 = 0.73$). Pairwise comparisons with Bonferroni adjustment showed that stride length was significantly greater at running intensities of 100% and 80% compared to 60% and 50% (Figure 2). The interaction between menstrual cycle phases and running intensities had no significant effect on stride length ($F_3 = 0.699$; $p = 0.471$, $\eta^2 = 0.091$).

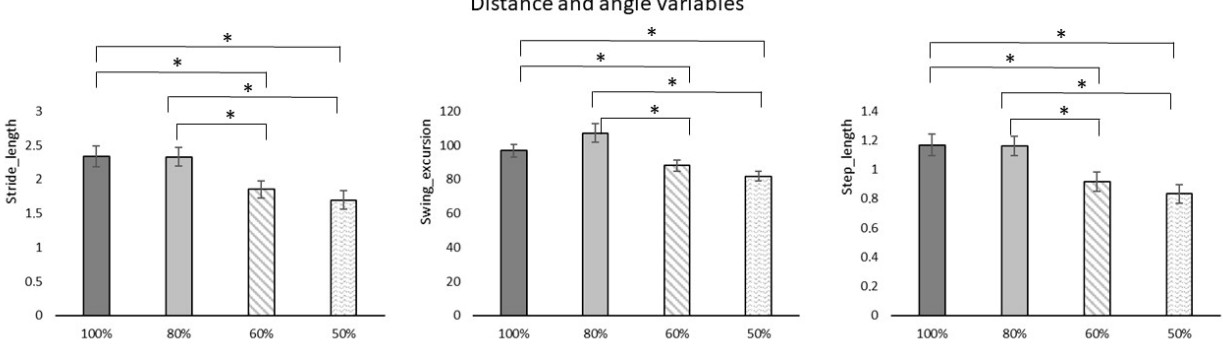

**Figure 2.** Comparison of the significant effect of test intensity on distance and angle variables. Stride_length (m); Swing_excursion (°); Step_length (m); * $p < 0.05$.

Table 2. Descriptions of running kinematics variables depending on the menstrual cycle phase at 50, 60, 80 and 100% of maximal velocity.

| | Intensity 50% Max_Velocity | | | Intensity 60% Max_Velocity | | | Intensity 80% Max_Velocity | | | Intensity 100% Max_Velocity | | |
|---|---|---|---|---|---|---|---|---|---|---|---|---|
| | Menstru | Follicular | Luteal | Menstru | Follicular | Luteal | Menstru | Follicular | Luteal | Menstru | Follicular | Luteal |
| **Kinematic Variables** | **M (SD)** | **M (SD)** | **M (SD)** | **M (SD)** | **M (SD)** | **M (SD)** | **M (SD)** | **M (SD)** | **M (SD)** | **M (SD)** | **M (SD)** | **M (SD)** |
| **Distances and angles** | | | | | | | | | | | | |
| Stride Length (m) | 1.70 (0.13) | 1.71 (0.17) | 1.68 (0.11) | 1.84 (0.14) | 1.88 (0.12) | 1.85 (0.12) | 2.28 (0.16) | 2.35 (0.13) | 2.36 (0.14) | 2.44 (0.3) | 2.14 (0.1) | 2.43 (0.25) |
| Stride Angle (°) | 1.5 (0.9) | 1.39 (1.1) | 1.65 (1.2) | 1.56 (1.1) | 1.49 (1) | 1.63 (1.2) | 1.95 (1.1) | 1.95 (0.9) | 2.05 (1.1) | 1.81 (1.1) | 1.8 (0.9) | 1.66 (0.9) |
| Swing Excursion (°) | 83.7 (10.6) | 80.3 (7.2) | 81.2 (11.7) | 89.1 (13.2) | 87.1 (7.3) | 88.3 (8.9) | 107 (18) | 106 (13) | 108.4 (16.5) | 98 (11.7) | 97.9 (11.5) | 94.4 (15.0) |
| Step Length (m) | 0.85 (0.07) | 0.83 (0.08) | 0.83 (0.05) | 0.91 (0.07) | 0.93 (0.07) | 0.91 (0.07) | 1.15 (0.8) | 1.16 (0.7) | 1.18 (0.7) | 1.21 (0.14) | 1.06 (0.05) | 1.23 (0.12) |
| **Velocity and frequency** | | | | | | | | | | | | |
| Velocidad máxima (km/h) | 7.56 (1.1) | 7.75 (0.8) | 7.56 (0.9) | 9.08 (1.3) | 9.3 (0.9) | 9.07 (1.1) | 12.1 (1.7) | 12.4 (1.2) | 12.1 (1.5) | 15.13 (2.1) | 15.5 (1.5) | 15.13 (1.9) |
| Max Pronation Velocity (°/s) | 505.5 (140.4) | 453 (190) | 507.5 (220) | 560 (171.4) | 570.1 (152.5) | 602.8 (187.2) | 733.8 (161) | 748 (209.6) | 769.1 (195.2) | 709.5 (175.3) | 625.6 (161.6) | 567.1 (247.3) |
| Max Stance Velocity (FS-MP) (°/s) | 726.9 (152.7) | 700.1 (263) | 713.1 (339.2) | 805.5 (244.1) | 811.3 (282.9) | 720.6 (428.2) | 981.7 (191.9) | 1052.5 (334) | 716.4 (405) | 875.8 (168) | 778.9 (191.4) | 829.6 (274.9) |
| Max Stance Velocity (MP-TO) (°/s) | 581.9 (97.3) | 588.4 (112.2) | 600 (130.5) | 612.1 (97) | 630.8 (65.7) | 632.2 (93.6) | 805.1 (157.7) | 836.5 (105.6) | 839.1 (148.7) | 757.9 (121) | 779.1 (132.2) | 769.3 (153.1) |
| Step rate | 164.9 (1.8) | 165 (2.5) | 161.9 (3.8) | 166 (2.5) | 168.1 (2.2) | 165.1 (2.6) | 172.8 (2.5) | 175.6 (1.9) | 172.4 (2.3) | 173 (2.5) | 173 (2.5) | 171 (2.6) |
| Vertical Speed (m/s) | −2.46 (0.47) | −1.8 (0.18) | −2.05 (0.09) | −2.65 (0.53) | −2.03 (0.13) | −2.29 (0.07) | −3.01 (0.47) | −2.48 (0.12) | −2.55 (0.13) | −3.14 (0.8) | −2.25 (0.07) | −3.44 (0.72) |
| **Times** | | | | | | | | | | | | |
| Contact Time (ms) | 329.9 (38.8) | 349.3 (72.2) | 346.4 (78.7) | 302.1 (40.9) | 293.8 (32.6) | 299.5 (35.3) | 262.5 (34.5) | 255.1 (26.9) | 259.8 (29.2) | 279.9 (41.5) | 279.9 (29.5) | 296.5 (36.7) |
| Contact Ratio (%) | 90.13 (11.4) | 95.4 (17) | 92.5 (16.5) | 83.5 (11.2) | 82.3 (9.2) | 82.4 (9.7) | 75.6 (10) | 74.6 (8) | 74.5 (8.9) | 79.8 (11.6) | 80.1 (9) | 83.1 (11.6) |
| Flight Time (ms) | 43.6 (43.9) | 19.4 (64) | 32 (62.1) | 60 (41.2) | 63.3 (33.8) | 64.6 (36) | 84.8 (36.2) | 86.5 (28.2) | 89 (32.1) | 71.8 (46.1) | 69.9 (34.7) | 63.8 (43.3) |
| Flight Ratio (%) | 13 (9.2) | 11.4 (11.3) | 13.4 (9.3) | 16.9 (10.5) | 17.8 (9.2) | 17.9 (9.4) | 24.5 (10) | 25.4 (8) | 25.4 (8.9) | 21.6 (10.2) | 21.4 (8.2) | 19.3 (9.3) |
| Time (MP-TO) (ms) | 279.5 (61.7) | 292.5 (65.6) | 303.8 (95.6) | 240.3 (39) | 241.9 (30.4) | 240.5 (33.7) | 207.4 (37.2) | 199.9 (29.7) | 204 (29.4) | 226.4 (44.5) | 221.3 (30.9) | 221.4 (40.8) |
| Time (TO-Min Swing) (ms) | 91.3 (47.1) | 69.6 (52.4) | 85.6 (54.1) | 115.3 (44) | 115.11 (33.7) | 120.1 (40.2) | 137.6 (39.7) | 135.4 (31.6) | 141.6 (35) | 118.3 (43.1) | 120.9 (39.5) | 109.4 (39.8) |
| Time (Max Swing-FS) (ms) | 78.6 (9.8) | 86.3 (5.9) | 91.9 (20.3) | 74.9 (9.2) | 83.3 (6) | 71.6 (8.3) | 104.8 (9.3) | 115.1 (7.7) | 101.5 (7.4) | 110.6 (10) | 106.9 (8.9) | 100.9 (8.6) |

Mentru—Menstruation phase; Max_stance_vel_MP—Maximum stance velocity from the point of maximum pronation to toe-off (°/s); Max_stance_vel_FS—Maximum stance velocity from the foot strike to the point of maximum pronation(°/s); Time_MP-TO—Time from the point of maximum pronation to toe-off (ms); Time_TO_min_swing—Time from toe-off to minimum swing point (ms); Time_TO_max_swing—Time from the maximum swing point to the foot strike (ms).

### 3.3.2. Stride Angle

The menstrual cycle phases had no significant effect on stride angle ($F_2$ = 0.262; $p$ = 0.773, $\eta^2$ = 0.036), which was slightly lower in the follicular phase (Table 2). Running intensity also had no significant effect on stride angle ($F_3$ = 1.53; $p$ = 0.257, $\eta^2$ = 0.179) (Figure 2). The interaction between menstrual cycle phases and running intensity had no significant effect on stride angle ($F_3$ = 0.410; $p$ = 0.674, $\eta^2$ = 0.055).

### 3.3.3. Swing Excursion

The menstrual cycle phases had no significant effect on swing excursion ($F_2$ = 0.692; $p$ = 0.517, $\eta^2$ = 0.090), which was slightly lower in the follicular phase, then in the luteal phase and finally reached its highest value in the menstruation phase (Table 2). Running intensity had a significant effect on swing excursion ($F_3$ = 27.8; $p$ < 0.001, $\eta^2$ = 0.80) (Figure 2). Pairwise comparisons with Bonferroni adjustment showed that Swing excursion was significantly higher at running intensities of 100% and 80% compared to 60% and 50% and tended to be higher at 100% than at 80%. The interaction between menstrual cycle phases and running intensity had no significant effect on swing excursion ($F_3$ = 0.398; $p$ = 0.724, $\eta^2$ = 0.054).

### 3.3.4. Step Length

The menstrual cycle phases had no significant effect on step length ($F_2$ = 0.675; $p$ = 0.525, $\eta^2$ = 0.088), which was slightly lower in the follicular phase (Table 2). Running intensity had a significant effect on step length ($F_3$ = 20.8; $p$ < 0.001, $\eta^2$ = 0.75). Pairwise comparisons with Bonferroni adjustment showed that step length was significantly higher at running intensities of 100% and 80% compared to 60% and 50% (Figure 2). The interaction between menstrual cycle phases and running intensities had no significant effect on step length ($F_3$ = 0.726; $p$ = 0.467, $\eta^2$ = 0.094).

### 3.4. Speeds and Frequencies

### 3.4.1. Maximum Pronation Velocity

The menstrual cycle phases had no significant effect on maximum pronation velocity ($F_2$ = 0.190; $p$ = 0.829; $\eta^2$ = 0.026), slightly lower in the follicular phase and higher in the menstruation phase (Table 2). Running intensity had a significant effect on maximum pronation velocity ($F_3$ = 14.3; $p$ < 0.001, $\eta^2$ = 0.67). Pairwise comparisons with Bonferroni adjustment showed that peak pronation velocity was significantly higher at 80% running intensity compared to 60% and 50% and also higher at 100% compared to 50% (Figure 3). The interaction between menstrual cycle phases and running intensities had no significant effect on maximum pronation velocity ($F_3$ = 0.257; $p$ = 0.096, $\eta^2$ = 0.269).

### 3.4.2. Maximum Stance Velocity from the Foot Strike to the Point of Maximum Pronation

The menstrual cycle phases had a significant effect on the maximum stance velocity from foot strike to the point of maximum pronation achieved in the test ($X_2^2$ = 9.2; $p$ = 0.010; $\eta^2$ = 0.138), with the median being significantly higher in the menstruation phase (Mn = 818; IQ = 285) than in the luteal phase (Mn = 706; IQ = 334) (r = 0.26; $p$ = 0.008) (Figure 1). Running intensity had a significant effect on the Maximum stance velocity from the foot strike to the point of maximum pronation reached in the test ($X_3^2$ = 36.0; $p$ < 0.001; $\eta^2$ = 0.325), with significant differences in all median pairwise comparisons, the highest being recorded at 80% (Mn = 916; IQ = 364), then at 100% (Mn = 804.5; IQ = 272), followed by 60% (Mn = 710; IQ = 237), and finally at 50% of running intensity (Mn = 639; IQ = 330) (Figure 3).

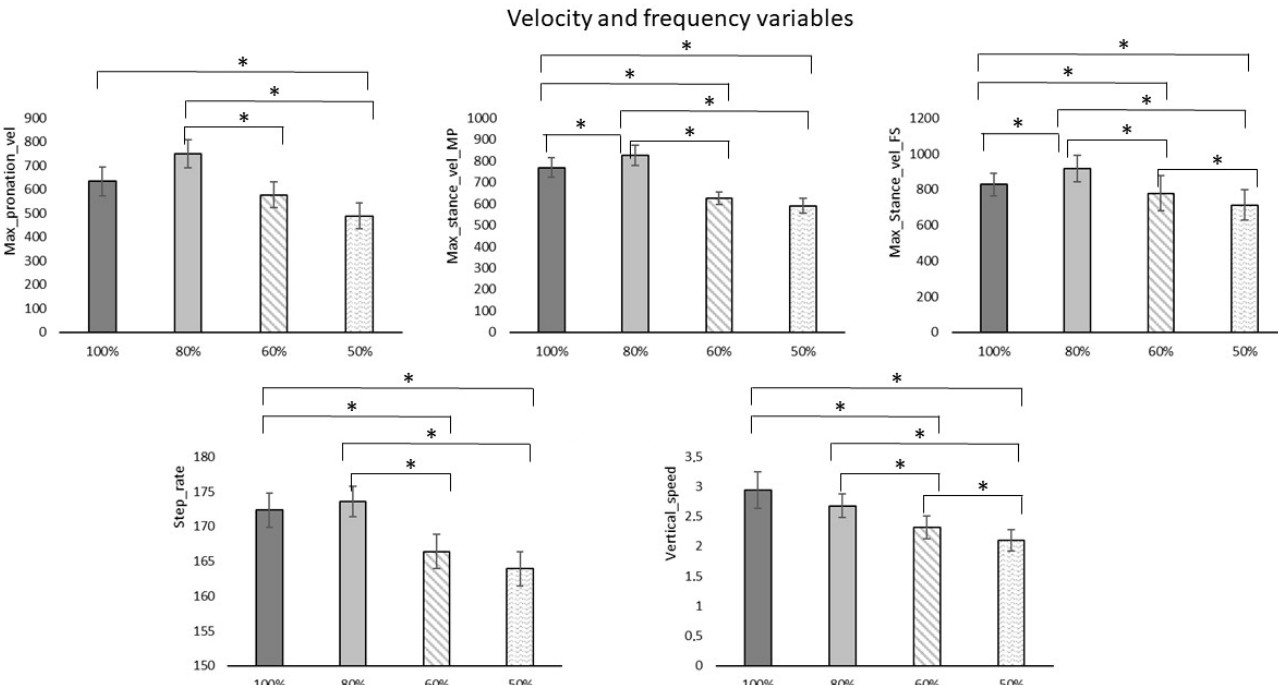

**Figure 3.** Comparison of the significant effect of test intensity on velocity and frequency variables. Max_pronation_vel—Maximum pronation velocity (°/s); Max_stance_vel_MP—Maximum stance velocity from the point of maximum pronation to toe-off (°/s); Max_stance_vel_FS—Maximum stance velocity from the foot strike to the point of maximum pronation (°/s); Step_rate (steps/min); Vertical speed (m/s); * $p < 0.05$.

### 3.4.3. Maximum Stance Velocity from the Point of Maximum Pronation to Toe-Off

The menstrual cycle phases had no significant effect on the maximum stance velocity from the point of maximum pronation to toe-off ($F_2 = 0.92$; $p = 0.421$, $\eta^2 = 0.116$), which was slightly lower in the menstruation phase (Table 2). Running intensity had a significant effect on the Maximum stance velocity from the point of maximum pronation to toe-off ($F_3 = 39.8$; $p < 0.001$, $\eta^2 = 0.85$). Pairwise comparisons with Bonferroni adjustment showed that maximum stance velocity from the point of maximum pronation to toe-off was significantly higher at 80% than at rest and at 100% than at 60% and 50% running intensity (Figure 3). The interaction between menstrual cycle phases and running intensity had no significant effect on maximum stance velocity from the point of maximum pronation to toe-off ($F_3 = 0.157$; $p = 0.892$, $\eta^2 = 0.022$).

### 3.4.4. Step Rate

The menstrual cycle phases had a significant effect on step rate ($F_2 = 5.48$; $p = 0.017$, $\eta^2 = 0.439$), which was significantly higher in the follicular phase than in the luteal phase (Table 2). Running intensity had a significant effect on step rate ($F_3 = 10.7$; $p = 0.008$, $\eta^2 = 0.60$). Pairwise comparisons with Bonferroni adjustment showed that the step rate was significantly higher at running intensities of 100% and 80% compared to 60% and 50% (Figure 3). The interaction between menstrual cycle phases and running intensity had no significant effect on step rate ($F_3 = 0.712$; $p = 0.505$, $\eta^2 = 0.092$).

### 3.4.5. Vertical Speed

The menstrual cycle phases had a significant influence on the vertical speed achieved in the test ($X_2^2 = 8.4$; $p = 0.015$; $\eta^2 = 0.116$), with the median being significantly lower in the follicular phase (Mn = −2.15; IQ = −0.6) than in the menstruation phase (Mn = −2.3; IQ = −0.75) (Z = −2.9; $p = 0.004$) and in the luteal phase (Mn = −2.3; IQ = −0.75) (Z = −2.9; $p = 0.003$) (Figure 1). Running intensity had a significant effect on the vertical speed

achieved in the test ($X_3^2$ = 38.0; $p < 0.001$; $\eta^2$ = 0.54) with significant differences in all median pairwise comparisons, except for the comparison between 80% and 100% of running intensity, with the highest value recorded at 80% (Mn = −2.5; IQ = −0.58) then at 100% (Mn = 2.35; IQ = −0.45), followed by 60% (Mn = −2.2; IQ = −0.45) and finally at 50% (Mn = −1.9; IQ = −0.47) (Figure 3).

### 3.5. Times and Percentages

#### 3.5.1. Contact Time

The menstrual cycle phases did not have a significant influence on contact time ($F_2$ = 1.87; $p = 0.192$, $\eta^2$ = 0.21), which was shorter in the luteal phase than in the menstruation phase and in the follicular phase, with very similar values (Table 2). Running intensity had a significant effect on contact time ($F_3$ = 15.6; $p = 0.003$, $\eta^2$ = 0.69). Pairwise comparisons with Bonferroni adjustment showed that contact time was significantly shorter at 80% running intensity than at the other intensities and shorter at 100% than at 50%. The interaction between menstrual cycle phases and running intensity had no significant effect on contact time ($F_3$ = 0.893; $p = 0.419$, $\eta^2$ = 0.113) (Figure 4).

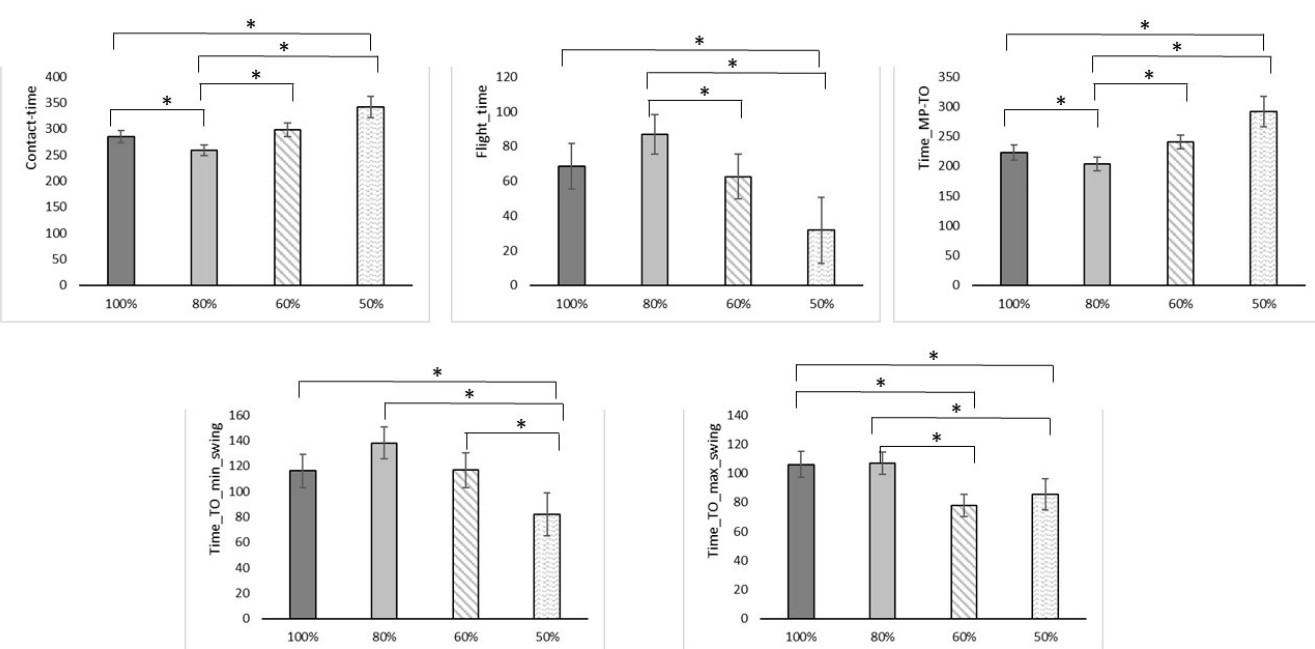

**Figure 4.** Comparison of the effect of test intensity on time variables. Contact_time (ms); Flight_time (ms); Time_MP-TO—Time from the point of maximum pronation to toe-off (ms); Time_TO_min_swing—Time from toe-off to minimum swing point (ms); Time_TO_max_swing—Time from the maximum swing point to the foot strike (ms); * = $p < 0.05$.

#### 3.5.2. Contact Ratio

The menstrual cycle phases had no significant effect on the contact ratio ($F_2$ = 0.82; $p = 0.46$, $\eta^2$ = 0.105), as all 3 phases had very similar values. Running intensity had a significant effect on the contact ratio ($F_3$ = 17.2; $p = 0.001$, $\eta^2$ = 0.71). Pairwise comparisons with Bonferroni adjustment showed that the contact ratio was significantly lower at 80% running intensity than at other intensities and lower at 100% than at 50%. The interaction between menstrual cycle phases and running intensity had no significant effect on the contact ratio ($F_3$ = 1.0; $p = 0.394$, $\eta^2$ = 0.125).

### 3.5.3. Flight Time

The menstrual cycle phases had no significant effect on flight time ($F_2 = 1.25$; $p = 0.317$, $\eta^2 = 0.15$), which was shorter in the follicular phase and higher in the menstruation phase (Table 2). Running intensity had a significant effect on Flight time ($F_3 = 11.4$; $p = 0.005$, $\eta^2 = 0.62$). Pairwise comparisons with Bonferroni adjustment showed that flight time was significantly higher at 80% running intensity than at 50% and 60% and higher at 100% than at 50% (Figure 4). The interaction between menstrual cycle phases and running intensities had no significant effect on Flight time ($F_3 = 1.1$; $p = 0.366$, $\eta^2 = 0.134$).

### 3.5.4. Flight Ratio

The menstrual cycle phases had no significant effect on the flight ratio ($F_2 = 1.30$; $p = 0.316$, $\eta^2 = 0.206$), which was higher in the luteal phase and lower in the menstruation phase (Table 2). Running intensity had a significant effect on the flight ratio ($F_3 = 26.0$; $p < 0.001$, $\eta^2 = 0.84$). Pairwise comparisons with Bonferroni adjustment showed that the flight ratio was significantly higher at 80% running intensity than at 50% and 60% and higher at 100% than at 50% (Figure 4). The interaction between menstrual cycle phases and running intensity had no significant effect on the flight ratio ($F_3 = 1.1$; $p = 0.364$, $\eta^2 = 0.180$).

### 3.5.5. Time from the Point of Maximum Pronation to Toe-Off

The menstrual cycle phases had no significant effect on the time from the point of maximum pronation to toe-off ($F_2 = 0.417$; $p = 0.667$, $\eta^2 = 0.056$), which was longer in the luteal phase compared to the menstruation phase and the follicular phase (both had very similar values) (Table 2). Running intensity had a significant effect on the time from the point of maximum pronation to take-off ($F_3 = 10.3$; $p = 0.013$, $\eta^2 = 0.60$). Pairwise comparisons with Bonferroni adjustment showed that the time from maximum pronation to toe-off was significantly shorter at 80% running intensity than at other intensities and shorter at 100% than at 50% (Figure 4). The interaction between menstrual cycle phases and running intensity had no significant effect on the time from maximum pronation to toe-off ($F_3 = 1.32$; $p = 0.298$, $\eta^2 = 0.158$).

### 3.5.6. Time from Toe-Off to Min Swing Point

The menstrual cycle phases had no significant effect on the time from toe-off to min swing point ($F_2 = 2.39$; $p = 0.160$, $\eta^2 = 0.255$), which was shorter in the follicular phase than in the menstruation and luteal phases (both with similar values) (Table 2). Running intensity had a significant effect on the time from toe-off to min swing point ($F_3 = 15.6$ $p < 0.001$; $\eta^2 = 0.69$). Pairwise comparisons with Bonferroni adjustment showed that the time from toe-off to min swing point was significantly shorter at 50% running intensity compared to the rest (Figure 4). The interaction between menstrual cycle phases and running intensities had no significant effect on the time from toe-off to min swing point ($F_3 = 1.33$; $p = 0.297$, $\eta^2 = 0.159$).

### 3.5.7. Time from the Max Swing Point to Foot Strike

The menstrual cycle phases had no significant effect on the time from max swing point to foot strike (ms) ($F_2 = 2.56$; $p = 0.113$, $\eta^2 = 0.267$), which was longer in the follicular phase than in the menstruation and luteal phases (both with similar values) (Table 2). Running intensity had a significant effect on the time from max swing point to foot strike ($F_3 = 9.4$; $p < 0.001$, $\eta^2 = 0.57$). Pairwise comparisons with Bonferroni adjustment showed that the time from max swing point to foot strike was significantly longer at 100% and 80% than at 50% and 60% running intensity (Figure 4). The interaction between menstrual cycle phases and running intensities had no significant effect on the time from max swing point to foot strike ($F_3 = 1.25$; $p = 0.310$, $\eta^2 = 0.152$).

## 4. Discussion

The aim of this study was to analyze the differences in running kinematic variables according to menstrual cycle phases and running intensity.

The main results showed that running intensity had a significant effect on all of these variables, with the exception of stride angle. Studies in the literature have also localized these findings, such as those of Orendurff et al. (2018) [4] and Albiach et al. (2021) [5]. The contact time was lower at 80% of the test intensity compared to 100%, 50% and 60%, and at 100% compared to 50%, the time the foot is on the ground will decrease due to the demands of increased running speed, and the need to quickly prepare the foot for the next landing in a continuous manner without losing speed. The stride length was greater at 100% and 80% intensity than at 60% and 50%; running speed is a function of both stride length and step rate; some runners increase running speed with an increase in stride length and others with an increase in step rate; therefore, the female runners in this study increased stride length with increasing running speed, and may tend to have higher ground impact forces and thus be more prone to possible lower limb injuries. The flight time was greater at 80% than at 50% and 60%, and greater at 100% of test intensity than at 50%; these findings were also found by Gijon-Nogueron et al. (2019) [6] in their study where they focused on female runners, and is related to the increase in stride length by increasing the time that a foot returns to contact with the ground. These same authors (Gijon-Nogueron et al., 2019 [6]) also reported similar results to this study in the maximum stance velocity from the point of maximum pronation to toe-off (propulsive phase), which also increases at 80% running intensity compared to 100%, 50% and 60%, and increases at 100% compared to 60% and 50%, which will be related to the contact time, both variables have the same behavior.

These results can give an orientation of the behavior of some kinematic variables in female runners, such as the increase in stride length and, therefore, the increase in flight time with increasing running speed; this will place the female runner with a higher probability of suffering an injury at higher speeds by increasing the impact forces against the ground due to this increase in stride length instead of increasing the step rate.

These kinematic variables will also be related to the phases of the menstrual cycle; the contact time decreases in the luteal phase compared to the menstruation and follicular phases, also taking into account that it decreases with increasing running speed, especially at 80%, it could be said that at 80% of maximum speed, women runners will be more efficient in the luteal phase. On the other hand, the data obtained from the step rate are contradictory, showing higher values in the follicular phase than in the luteal phase, higher efficiency in the follicular phase, and an increase in running speed. (100% and 80%). These two kinematic variables related to efficiency are reflected in the flight ratio, which is higher in the luteal phase and lower in the menstruation phase, as well as increasing with running speed, especially at 80%. High flight ratio values indicate greater running efficiency; therefore, with this data, we can resolve the doubt raised, indicating that women runners at 80% of their maximum speed are more efficient in the luteal phase, whereas Bingzheng et al. (2023) [21] in their study placed the mid-luteal phase with the lowest risk of injury.

Winter et al. (2020) [13] and Popp et al. (2023) [18] show in their studies that all variables related to vertical load in female runners will also be potential indicators of injury risk; in this study the vertical speed variable was higher at 80%, then 100%, 60%, and finally 50%, and was lower in the follicular phase than in the menstruation and luteal phases, these data are related to those discussed in previous paragraphs, relating an increase in stride length to an increase in running speed, associated with an increased risk of injury due to impact forces, which was slightly lower in the follicular phase, therefore, in the follicular phase there could be a lower risk of injury in female runners, and as for the highest risk of injury as in the study by Bell et al. (2014) [20], it could be in the menstruation phase.

According to Wilzman et al. (2022) [19], increased pronation may be another cause of injury, obtaining the same results as in this study where the maximum pronation velocity was higher at 80% and 100% running intensity; therefore, as Winter et al. (2020) [13] stated

in their study that training at high speeds for female runners may be one of the causes of injury risk.

Although the results of this study are applicable for adjusting the type of training according to menstrual phase and running intensity, they also have some limitations. In this case, the main limitation is that the sample is small and future studies should consider women with different characteristics, in terms of age, lifestyle or sport practiced.

**5. Conclusions**

This study presents novel findings regarding the relationship between kinematic data, the menstrual cycle, and performance in female athletes. The results demonstrate that certain kinematic variables can vary depending on the menstrual cycle, and some of these variations can influence performance, relating to possible injury risk factors in previous literature. These findings have important practical implications for female athletes related to the menstrual cycle and identify changes in performance associated with running technique and related to possible injury risk.

The data relating to running efficiency is the flight ratio, formed by the variables of contact time and step rate; in this study, women are more efficient in the luteal phase at 80% of maximum speed, presenting a lower efficiency in the menstruation phase.

The data relating to the vertical speed and the maximum pronation velocity are related to a higher risk of injury; in this study, women at intensities of 80% and 100% of the maximum velocity presented higher risks of injury. In addition, the vertical speed is higher in the menstruation phase, giving it a higher risk of injury, and is lower in the follicular phase, giving it a lower risk of injury.

These results suggest that training at high intensities could be a factor of greater risk of injury in female athletes, especially in the menstruation phase, finding in the luteal phase and at an intensity of 80% greater efficiency in the running, taking into account that in the follicular phase could be located a lower risk of injury.

**Author Contributions:** Conceptualization, C.D.-M., J.d.C. and A.G.; methodology, C.D.-M., J.d.C. and A.G.; formal analysis, C.D.-M., J.R.-L., J.G. and R.M.-G.; data curation, C.D.-M., J.R.-L., J.G. and R.M.-G.; writing—original draft preparation, C.D.-M., J.d.C. and A.G.; writing—review and editing, C.D.-M., J.R.-L., J.G. and R.M.-G. All authors have read and agreed to the published version of the manuscript.

**Funding:** This research received no external funding.

**Institutional Review Board Statement:** The study was conducted in accordance with the Declaration of Helsinki, and approved by the Research Ethics Committee of the Autonomous University of Madrid, protocol code 51401469B (16 July 2021).

**Informed Consent Statement:** Informed consent was obtained from all subjects involved in the study.

**Data Availability Statement:** The data presented in this study are available on request from the corresponding author.

**Conflicts of Interest:** The authors declare no potential conflicts of interest with respect to the research, authorship, and/or publication of this article.

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
