# Peer review of "Analysis of Kinematic Variables According to Menstrual Cycle Phase and Running Intensity: Implications for Training Female Athletes"

_applsci, doi:10.3390/app14125348_

Round 1

Reviewer 1 Report

Comments and Suggestions for Authors

In the abstract be sure to denote what was your values for significance and put in some of the results values as well. Also what stat tests that you used.

The opening paragraph in the introduction can be moved to later on. In paragraph 5 there is some awkward phrasing and change your use of "however"

In the methods what time of day did you test? did you make that consistent for each phase with the same sleep, nutrition, and caffeine intake? likely replace the " ' " for "min". change cannot to "can't". Make sure your writing is in the past tense since you have completed all of this work. 

On page 4 why does your formatting shift suddenly?

final line of the statistical analysis is awkward and needs to be reworked. 

With your significant values, what are the effect sizes? be sure to include these as well. Your writing is quite repetitive in sections make sure there is a bit more difference between each paragraph in page 5.

on page 6 further establish with your data how the cycle phases tended to influence the contact time.

fix the titles of your figures and tables to be accurate and shorten the tables to fit on the page without so much dead space on them. On your figures be sure to take the y axis through 0 to make the differences better contextualized. Don't use "a" and "b" for significance, instead use "*" and the dagger option. 

In the discussion condense the 1st and 2nd paragraph, your comparison to values are odd in that the 100% level was significantly lower then the 80%?

Try to give more context on the magnitude and perhaps the percentage changes exhibited over the different testing times. 

Comments on the Quality of English Language

The paper is well written and doesn't need much modification.

Author Response

Response to Reviewer 1 Comments

1. Abstract.

In the abstract be sure to denote what was your values for significance and put in some of the results values as well. Also what stat tests that you used.

Thank you very much for this comment. We have added the significance values and we have indicated the statistical tests used.

2. Introduction.

The opening paragraph in the introduction can be moved to later on. In paragraph 5 there is some awkward phrasing and change your use of “however"

Following your instructions, we have placed the first paragraph at the end of the introduction, as a closing paragraph.

We have also modified the fifth paragraph to make it easier to read.

3. Methods.

In the methods what time of day did you test? did you make that consistent for each phase with the same sleep, nutrition, and caffeine intake? likely replace the " ' " for "min". change cannot to "can't". Make sure your writing is in the past tense since you have completed all of this work. 

Thank you for these points.

The test appointments used to be in the same time slot.

It is explained in the text as follows: Participants were instructed to maintain their regular diet and abstain from vigorous physical activity for 24-48 hours before the test. They were also advised to avoid consuming food, stimulants, or ergogenic aids for at least three hours prior to the test.

Verbs that were not in the past tense have been modified.

4. Statistical analysis.

Final line of the statistical analysis is awkward and needs to be reworked. 

We have rewritten the sentence to: “Statistical significance was adjusted for p-values lower than 0.05”

5. Results.

With your significant values, what are the effect sizes? be sure to include these as well.

Thank you. When indicating in ANOVA tests we did not want to incorporate Cohen's d in pairwise comparisons because the results would become too cumbersome, and eta squared serves as the effect size statistic of the ANOVA test.

As effect size statistics we used eta squared with values of 0.01 small, 0.06 medium and 0.14 large.

For the nonparametric tests we used the r statistic, with values of 0.5 large, 0.3 medium and 0.3 small. This information has been added in the statistical analysis section. Reference also included:

Fritz, C. O., Morris, P. E., & Richler, J. J. (2012). Effect size estimates: current use, calculations, and interpretation. Journal of experimental psychology: General, 141(1), 2.

Your writing is quite repetitive in sections make sure there is a bit more difference between each paragraph in page 5.

In the presentation of the results to facilitate understanding we have tried to simplify and be very systematic in the wording (O'donoghue, 2009), so that we always comment first on the effects of menstrual cycle phases, then comment on the effects of career intentionality and finally, on the interaction of both variables.

O'donoghue, P. (2009). Research methods for sports performance analysis. Routledge.

6. Figures and tables.

Fix the titles of your figures and tables to be accurate and shorten the tables to fit on the page without so much dead space on them. On your figures be sure to take the y axis through 0 to make the differences better contextualized. Don't use "a" and "b" for significance, instead use "*" and the dagger option.

We have made the proposed changes both in figures and in tables.

7. Discussion.

In the discussion condense the 1st and 2nd paragraph, your comparison to values are odd in that the 100% level was significantly lower then the 80%?

We have not understood how to modify this suggestion.

Reviewer 2 Report

Comments and Suggestions for Authors

Introduction

fifth paragraph- translate spanish oration

sixth paragraph- last sentences without reference

Although only some studies have analysed one or another kinematic variable, it would be worth mention them (e.g., Williams, T. J., & Krahenbuhl, G. S. (1997). Menstrual cycle phase and running economy. Medicine and science in sports and exercise29, 1609-1618), and also, there are more general approaches, like reviews, that deserve to be mentioned (e.g., McNulty, K. L., Elliott-Sale, K. J., Dolan, E., Swinton, P. A., Ansdell, P., Goodall, S., ... & Hicks, K. M. (2020). The effects of menstrual cycle phase on exercise performance in eumenorrheic women: a systematic review and meta-analysis. Sports medicine50, 1813-1827.); in order to reinforce your findings.

Subjects

Include code number identification of the approval from the Research Ethics Committee

Design

Identify model of shoes used and conditions (new or old)

Materials and Measurements

Identify references used to define maximal and submaximal tests

Statistical Analysis

For correlations include confidence intervals, keeping respective signals (mandatory)

For Friedman and Wilcoxon identify and include effect size estimation (mandatory)

Tables 2-5

- review legends (you have no corelations, only descriptive statistics)

- DT is SD?

- during Follicular phase most of the SD suffer a strong alteration. we suggest that you discuss this pattern.

Discussion

page 13, third paragraph, first line- "will also be" or are...?

last paragraph- translation from spanish not completed (is the second one, review all document)

Include limitations of the study (e.g., sample dimension)

Comments on the Quality of English Language

Some phrases are still in spanish. 

Author Response

Response to Reviewer 2 Comments

  1. Introduction

Fifth paragraph- translate spanish oration

Sixth paragraph- last sentences without reference

Although only some studies have analysed one or another kinematic variable, it would be worth mention them (e.g., Williams, T. J., & Krahenbuhl, G. S. (1997). Menstrual cycle phase and running economy. Medicine and science in sports and exercise29, 1609-1618), and also, there are more general approaches, like reviews, that deserve to be mentioned (e.g., McNulty, K. L., Elliott-Sale, K. J., Dolan, E., Swinton, P. A., Ansdell, P., Goodall, S., ... & Hicks, K. M. (2020). The effects of menstrual cycle phase on exercise performance in eumenorrheic women: a systematic review and meta-analysis. Sports medicine50, 1813-1827.); in order to reinforce your findings.

Thank you for these comments.

We have translated the fifth paragraph into English.

The sixth paragraph is all referenced, the last part is referenced by the following author Wilzman et al. (2022).

In the research of Williams T. J. and Krahenbuhl G. S. (1997) we have not found the study of any kinematic variable that could help us to reference our results.

In the research by McNulty et al (2020) we have not found any specific data regarding our research that could help us to reference our results.

2. Subjects.

Include code number identification of the approval from the Research Ethics Committee

This article is part of a doctoral thesis that will be presented in the coming months (once it has been accepted). Before carrying out the research, it was submitted to the Ethics Committee of the Universidad Autónoma de Madrid, where we were provided with the attached document:

Ethics Committee of the Universidad Autónoma de Madrid

3. Design.

Identify model of shoes used and conditions (new or old)

This is a very good consideration.

The women tested wore their usual training or competition shoes.

3. Materials and Measurements

Identify references used to define maximal and submaximal tests.

We have referenced these tests with some articles we already had in our bibliography.

4. Statistical Analysis

For correlations include confidence intervals, keeping respective signals (mandatory).

We have added confidence intervals in Table 1

For Friedman and Wilcoxon identify and include effect size estimation (mandatory).

As effect size statistics for Friedman tests we used eta squared with values of 0.01 small, 0.06 medium and 0.14 large.

For Wilcoxon tests we used the r statistic, with values of 0.5 large, 0.3 medium and 0.3 small.

This information has been added in the statistical analysis section. Reference also included:

Fritz, C. O., Morris, P. E., & Richler, J. J. (2012). Effect size estimates: current use, calculations, and interpretation. Journal of experimental psychology: General, 141(1), 2.

We did not want to incorporate Cohen's r in pairwise comparisons because the results would become too cumbersome, and eta squared serves as the effect size statistic of the Friedman test.

5. Tables 2-5

Review legends (you have no corelations, only descriptive statistics)

Thank you. We deleted Correlations.

DT is SD?

during Follicular phase most of the SD suffer a strong alteration. we suggest that you discuss this pattern.

We changed SD instead DT.

6. Discussion

Page 13, third paragraph, first line- "will also be" or are...?

Last paragraph- translation from spanish not completed (is the second one, review all document)

Include limitations of the study (e.g., sample dimension)

We have chosen ‘will also be’ which is perhaps most appropriate for reading comprehension.

English translations have been made.

At the end of the discussion we have added some limitations of the study and future lines of research.

Reviewer 3 Report

Comments and Suggestions for Authors

Review on analysis of kinematic variables according to menstrual cycle phase and running intensity: implications for training female athletes

General comments

The study investigates the effect of maturational cycle on running kinematics in female athletes, which is a very interesting topic. However, I have some concerns about the study. Firstly the study only investigated 8 subjects, which is a bit low to do proper statistics upon. That is also why not many differences were found due to the menstrual cycle. This makes it very difficult to come with some conclusions based upon the findings. Secondly, the study describes things that go beyond the findings, like less injuries in the conclusion. Try to stick to the findings of the study and not speculate on a basis you have not found. Thirdly, some sentences are still in Spanish in the article. Fourthly, there is no specific research question postulated in the introduction, which makes it difficult for the readers to know what you exactly are going to study. Include also a hypothesis with reference or theory about what you expect as outcome and why exactly that.

The introduction is way too long and has information which is not so important for the research question. Especially the part about injury risk reduction. As you are only going to investigate the effect of Menstrual cycle the part of injury risk reduction is redundant and not interesting for the study. Suggest to delete or only writ one or two sentences about this.

In the methods part it is not clear how different parameters were measured and why exactly these parameters. Where the parameters measured with two IMUs and what was sampling rate and why parameters like pronation, and not dorsal flexion. What is vertical feet of foot when it is in stance phase? Pretty low I would suggest as it is in stance on the ground or is this the dorsal flexion and pronation movement during stance.

Why measuring flight ratio, when you already have contact ratio, which is the opposite of the flight ratio.

In methods part under speeds and frequencies the first part (The start of the test …. Hour). is repetition of what is mentioned before. Suggest to delete it in one of the two places.

In statistical analysis correlations with maximal speed were calculated. If they were not significant, they were not investigated further. However, in results part all were significantly correlated, so what is the point then with this analysis.

The results part is way too long and thereby not easy to follow for the reader. Firstly the results are shown in figures and the same results again in tables. That is double reporting, which should be avoided. Suggest to make one table of table 2, 3 and 4. That is enough.

Furthermore, reporting all the parameters individually makes it difficult to follow. You can just take together all the significant menstrual effects on the different parameters and wrote these were significantly affected by the menstrual cycle, while the others were not. Furthermore, that differences in running kinematics were found between the different running intensities is not one of the research aims. So why so much focus on this running intensity. This could also be written shorter in the results part by taking together several parameters that were affected and write this up once.

In discussion the discussion starts with differences in kinematics in running intensities, which are not the main focus (according to the title of the study). You should start with discussing the findings of the menstrual cycle on kinematics and discuss why exactly these findings with some theoretical explanations. The part of differences in kinematics due to running intensities are not interesting, thereby not much to discuss about. Now the main part of discussion is on the wrong findings. So change this.

The conclusions are not related so much to the title of the study, as mentioned about the discussion. Furthermore, it is related to injury risk, which you have not investigated and thereby , you cannot make  those statements as you do later in the conclusion part. The conclusion part can only refer to the main findings with an evt. explanation for this. Suggest to delete the rest.

Comments on the Quality of English Language

Some sentences are still in Spanish, which is difficult to unerstand, when reading English

Author Response

Response to Reviewer 3 Comments

  1. General comments.

The study investigates the effect of maturational cycle on running kinematics in female athletes, which is a very interesting topic. However, I have some concerns about the study. Firstly the study only investigated 8 subjects, which is a bit low to do proper statistics upon. That is also why not many differences were found due to the menstrual cycle. This makes it very difficult to come with some conclusions based upon the findings. Secondly, the study describes things that go beyond the findings, like less injuries in the conclusion. Try to stick to the findings of the study and not speculate on a basis you have not found. Thirdly, some sentences are still in Spanish in the article. Fourthly, there is no specific research question postulated in the introduction, which makes it difficult for the readers to know what you exactly are going to study. Include also a hypothesis with reference or theory about what you expect as outcome and why exactly that.

One of the main limitations of this study was the complexity of measuring women on the exact days of each phase, coinciding with the days and times they were available and free from work, family, and other personal commitments. This limited the number of participants to eight. Thus, the effect size in various statistical tests was medium, but possibly no significant differences were found due to a type 2 statistical error, and it would be very useful to be able to increase the number of participants in future research.

There are several articles, cited in the study, that relate the biomechanics of running to the production of injuries.

We have translated into English.

The aim of this study is to analyse whether there are changes in the kinematic variables in running throughout the menstrual cycle and to relate them to running performance and injury prevention.

2. Introduction.

The introduction is way too long and has information which is not so important for the research question. Especially the part about injury risk reduction. As you are only going to investigate the effect of Menstrual cycle the part of injury risk reduction is redundant and not interesting for the study. Suggest to delete or only writ one or two sentences about this.

There are several articles, cited in the study, that relate the biomechanics of running to the production of injuries, and some of these articles link the possibility of an increased risk of injury to some phases of the menstrual cycle.

3. Methods.

In the methods part it is not clear how different parameters were measured and why exactly these parameters. Where the parameters measured with two IMUs and what was sampling rate and why parameters like pronation, and not dorsal flexion. What is vertical feet of foot when it is in stance phase? Pretty low I would suggest as it is in stance on the ground or is this the dorsal flexion and pronation movement during stance.

Why measuring flight ratio, when you already have contact ratio, which is the opposite of the flight ratio.

In methods part under speeds and frequencies the first part (The start of the test …. Hour). is repetition of what is mentioned before. Suggest to delete it in one of the two places.

The variables studied are those measured by RunScribe, and each one of them has been explained for a better understanding, as well as relating them to previous studies by other authors.

4. Statistical Analysis.

In statistical analysis correlations with maximal speed were calculated. If they were not significant, they were not investigated further. However, in results part all were significantly correlated, so what is the point then with this analysis.

Thanks for your comments. When selecting which kinematic variables provided by Runscribe, we only estimated those that had a significant correlation with the maximum velocity. Therefore, all the variables that were analyzed in the study have a significant correlation with the maximum speed of the test.

5. Results.

The results part is way too long and thereby not easy to follow for the reader.

We understand your comment. It is normal that it is long, as it analyzes 17 dependent variables structured in 3 sections (distance and angles, velocities and frequencies, and times) in relation to 2 independent variables and their interaction. We think that we have synthesized well how each of the independent variables affects each dependent variable in a single paragraph.

Firstly the results are shown in figures and the same results again in tables. That is double reporting, which should be avoided. Suggest to make one table of table 2, 3 and 4. That is enough.

In reality, the tables represent the descriptive data with specific values, while the figures represent the significant differences of the pairwise comparisons with no specific values for descriptive data.

Furthermore, reporting all the parameters individually makes it difficult to follow. You can just take together all the significant menstrual effects on the different parameters and wrote these were significantly affected by the menstrual cycle, while the others were not. Furthermore, that differences in running kinematics were found between the different running intensities is not one of the research aims. So why so much focus on this running intensity. This could also be written shorter in the results part by taking together several parameters that were affected and write this up once.

Thank you for your comment. When structuring the results, we thought of two options. The first one was to structure them on the basis of the independent variables, as you propose. In this case, since there are 17 dependent variables, it would be very difficult to find the specific results for a particular dependent variable. For this reason, we decided to structure based on the dependent variables, instead of the independent variables.

Regarding to the aim of the study we have added “running intensity” as follows: “This research aims to identify the running technique variables related to performance and influenced by menstrual cycle and running intensity.”

6. Discussion.

In discussion the discussion starts with differences in kinematics in running intensities, which are not the main focus (according to the title of the study). You should start with discussing the findings of the menstrual cycle on kinematics and discuss why exactly these findings with some theoretical explanations. The part of differences in kinematics due to running intensities are not interesting, thereby not much to discuss about. Now the main part of discussion is on the wrong findings. So change this.

In the title of the article, the running intensity is mentioned, so it will be an important piece of information for us in the research.

7. Conclusions.

The conclusions are not related so much to the title of the study, as mentioned about the discussion. Furthermore, it is related to injury risk, which you have not investigated and thereby , you cannot make  those statements as you do later in the conclusion part. The conclusion part can only refer to the main findings with an evt. explanation for this. Suggest to delete the rest.

Thanks for your comment. We agree. In the conclusion we reported about objective results according to the aims of the study in the second and third paragraph. In the fourth paragraph we summary “possible suggestions” of the results. As this fourth paragraph is not an objective information we reported that “results suggest…”. Thus, the data on the possible risk of injury has been given to us by previous research articles by other authors mentioned in our article.

Round 2

Reviewer 2 Report

Comments and Suggestions for Authors

No further comments

Author Response

Thank you very much for all your contributions to improve our research paper.

Reviewer 3 Report

Comments and Suggestions for Authors

The introduction and results part are still too long with too much repetition ofr figures and tables. This makes it difficult to follow the flow of the study. Furthermore 8 subjects are not enough to come to some good conclusions.

Comments on the Quality of English Language

There are still some small errors in the english language, which have to be looked at.

Author Response

Dear reviewer.

The article fits the word count of the journal because it is 5383 words excluding abstract, references, tables and figures.

The introduction has been increased to 956 words among other reasons to accommodate requests from two other reviewers. Although we like the introduction as it is now, and two other reviewers have also approved the introduction as it is now, if you are able to justify to us what part of the information in the introduction is left over, we will try to accommodate your specific proposals.

As for the results, to conform to your guidelines we have reduced tables 2,3,4 and 5 to a single table. For us there are 3 arguments to justify not repeating the information in the tables and figures as follows:

1- A reader looking at the tables can see precise descriptive data, but cannot know if there are significant differences, nor between which groups of the independent variables.

2- A reader looking at the figures can know if there are significant differences and between which groups of the independent variables, but cannot know specific descriptive data.

3- A person looking at the tables and figures cannot know what the precise statistics of the main effect of the independent variables are unless he or she reads the wording of the results.

On the other hand, in terms of the structure of presentation of the results, a reader who wants to look for the precise results of how the independent variables affect a particular variable will find it very easy to find them by structuring the wording of the results based on the dependent variables. In addition, a reader who wishes to know in a summarized way the results of each independent variable on the dependent variables need only go to the figures, since the figures have been structured on the basis of the independent variables.

We agree with the reviewer that eight subjects are too few participants to extrapolate to women's populations in general, despite the difficulty of accessing the sample and making 96 records. We have therefore incorporated sample size as possible limitations of the study by adding the following paragraph at the end of the discussion:

"Although the results of this study have an application to adjust the type of workouts according to the phase of menstruation and running intensity, they also present some limitations. In this case, the main limitation is that the sample is small and future studies should consider women with different characteristics, in terms of age, life habits or sport practiced."
